# Metabolic Therapy of Heart Failure: Is There a Future for B Vitamins?

**DOI:** 10.3390/ijms23010030

**Published:** 2021-12-21

**Authors:** Jérôme Piquereau, Solène E. Boitard, Renée Ventura-Clapier, Mathias Mericskay

**Affiliations:** UMR-S 1180, Inserm Unit of Signaling and Cardiovascular Pathophysiology, Faculty of Pharmacy, Université Paris-Saclay, 92296 Chatenay-Malabry, France; Solene.boitard@inserm.fr (S.E.B.); renee.ventura@universite-paris-saclay.fr (R.V.-C.)

**Keywords:** heart failure, energy metabolism, mitochondria, B vitamins, thiamin, riboflavin, nicotinamide, pyridoxine, folate, cobalamin, metabolic therapy

## Abstract

Heart failure (HF) is a plague of the aging population in industrialized countries that continues to cause many deaths despite intensive research into more effective treatments. Although the therapeutic arsenal to face heart failure has been expanding, the relatively short life expectancy of HF patients is pushing towards novel therapeutic strategies. Heart failure is associated with drastic metabolic disorders, including severe myocardial mitochondrial dysfunction and systemic nutrient deprivation secondary to severe cardiac dysfunction. To date, no effective therapy has been developed to restore the cardiac energy metabolism of the failing myocardium, mainly due to the metabolic complexity and intertwining of the involved processes. Recent years have witnessed a growing scientific interest in natural molecules that play a pivotal role in energy metabolism with promising therapeutic effects against heart failure. Among these molecules, B vitamins are a class of water soluble vitamins that are directly involved in energy metabolism and are of particular interest since they are intimately linked to energy metabolism and HF patients are often B vitamin deficient. This review aims at assessing the value of B vitamin supplementation in the treatment of heart failure.

## 1. Introduction

Heart failure (HF), as the outcome of many cardiovascular diseases, is one of the major causes of death in industrialized countries and a substantial economic burden worldwide. According to recent estimations, 1 to 2% of the global adult population could be affected by this syndrome, with a prevalence of approximately 65 million patients worldwide [1]. This figure is expected to increase as the population ages, urging the scientific community to expand their therapeutic arsenal to improve the survival rate of HF patients. Although intensive research in the last decades has improved our understanding of HF pathophysiology and the means to tackle it, the 5-year survival rate of 55 to 60% remains unsatisfying [2]. Standard HF therapies target the dysregulation of the neurohormonal system or reduce the cardiac workload. These therapies, however, neglect the profound alterations of energy metabolism encountered in HF due to the lack of efficient tools for the metabolic treatment of this syndrome. Current knowledge suggests that new therapies aiming at restoring the energy balance in the heart should be used on top of current therapies to better treat HF patients and improve quality of life and possibly survival.

The heart is one of the main energy consumers of the body owing to its incessant and rhythmic blood pumping activity throughout the circulatory system. Cardiac contractile properties rely on molecular mechanisms involving ATPases (sarcoplasmic/endoplasmic reticulum Ca2+ ATPase and myosin ATPase especially) that require suitable energy production since cardiomyocytes stocks of high-energy phosphate molecules; i.e., adenosine triphosphate (ATP) and phosphocreatine (PCr), can only ensure cardiac contraction for a few seconds only. This limitation imposes the requirement of the constant production of energy molecules to adapt to the fluctuating cardiac demand. The heart is thus endowed with impressive ATP production capacities depending on tightly regulated metabolic processes, the alterations of which have extensively been described in HF settings since the 1980s [3]. It is now clear that the failing heart can be considered as an engine unable to properly oxidize fuel and sustain its own energy demand with 25–35% lower ATP content and 40–50% lower PCr content than a healthy heart [4,5,6]. This incapacity is due to profound perturbations in the regulations and the efficiency of many reactions/processes involved in energy molecule production [7]. Despite intensive work in this field, identifying novel therapeutic strategies remain challenging. Cardiac energy metabolism exhibits a high complexity involving many interlaced signaling pathways. Any metabolic therapy aiming at restoring the energetic disorders of the failing heart necessitates the development of global strategies that would impact all the actors of the energy machinery of the cardiomyocyte. Numerous recent strategies using synthetic chemical compounds to target specific aspects of energy metabolism have been proposed and tested with mitigated success so far [8,9], prompting the search for new compelling alternatives.

Natural compounds, including vitamins, are increasingly gaining scientific attention due to their significant involvement in numerous elementary reactions of cellular homeostasis [10]. Vitamin supplementations have already been part of treatments of specific diseases for many years [11,12] and have also been tested in the frame of HF with variable degrees of success [13]. However, conflicting results from a series of small clinical studies [14,15,16,17,18,19,20] only showed that more research is needed to better define which type of vitamin supplementation protocol should be implemented and which HF etiologies and stages of the syndrome should selectively be targeted by these potential treatments. To face the profound metabolic disorders described in HF, the big family of B vitamins are emerging as promising supplements to manage HF for two main reasons; first, B vitamins are highly involved in numerous elementary processes of cardiac energy metabolism [21] and second, B vitamins are deficient in HF patients due to the use of diuretics, malnutrition, and advanced age [22,23]. Vitamins are precursors of essential coenzymes playing a major role in energy metabolism. In this review, we describe the main features of cardiac energy metabolism in the healthy heart and its abnormalities in HF with emphasis on the role of B vitamins. We then review the use of B vitamins in the preclinical and clinical context of HF while assessing the value of B vitamin supplementation in the treatment of this debilitating syndrome.

## 2. The Heart: An Oxidative Tissue

The heart never stops beating from early in utero life and throughout the lifetime. Cardiac function demands a large amount of energy, cycling approximately 6 kg of ATP each day [24]. The healthy mature heart obtains energy by metabolizing a large variety of energy substrates depending on substrate availability and/or energy demand. For that reason, the heart is qualified as a metabolic omnivore. Under normal conditions, energy molecules are mainly (60% to 90%) generated from the metabolism of fatty acids (FA), while the remaining molecules are produced from carbohydrates through pyruvate oxidation [7,9,25]. Under specific conditions, such as long fasting, the heart resorts to alternative energy producing substrates, including lactate, ketone bodies, and amino acids, especially branched-chain amino acids (BCAA). Cardiomyocytes are thus equipped with a whole range of substrate-specific membrane transporters, the expression of which is modulated according to the cardiac environment [26,27]. As the heart exhibits a low storage ability, they allow the constant entry of these substrates to supply the energy machinery. The production of ATP within the cardiomyocyte strongly relies on the mitochondrial oxidative function, which provides 90% to 95% of energy molecules from the oxidation of the aforementioned substrates [7,28]. The remaining 5% to 10% are produced by the anaerobic glycolysis pathway, degrading glucose to pyruvate. However, pyruvate can enter mitochondria for further catabolism processes, in which case the full process, cytosolic glycolysis and pyruvate oxidation, is called glucose oxidation. The flux of a given substrate to oxidative pathways is largely dependent on the expression and the activity of rate-limiting enzymes that are respectively modulated by transcriptional and post-translational processes, which are under the control of many intertwined signaling pathways including pathways regulated by vitamins, as will be detailed later. In the end, the orientation of the cardiac metabolism is the result of intricate machinery in which the availability of the substrates itself plays a key role. For instance, the oxidation of a specific substrate can favor the mechanisms responsible for its degradation while repressing the utilization of any other potential energy sources. This phenomenon involves feedback mechanisms known as the Randle cycle [29], in which FA oxidation inhibits glucose utilization and vice versa, underlying the flexibility of cellular metabolism that is able to adapt to the changes of the cell environment.

Regardless of the type of recruited substrates, their oxidation consists of producing reducing agents (NADH and FADH_2_) that will then act as electron donors to the mitochondrial electron transfer chain (ETC). The energy resulting from electron flow within this chain is coupled to proton translocation from the matrix to the intermembrane space of the mitochondria. Thus, the energy produced from substrate oxidation is transformed into a chemical and electrical gradient, the proton motive force, which in turn is used by the F_O_/F_1_ ATP synthase of the mitochondrial inner membrane to rephosphorylate ADP [30]. ATP production is dependent on the return of the protons accumulated in the intermembrane space to the matrix through the F_O_ portion of the ATP synthase, inducing the rotation of the γ subunit of the F_1_ portion [31]. As a consequence, the proton gradient generated by the ETC is consumed as the ATP is produced and has to be constantly re-established using reduced coenzymes, NADH, and FADH_2_ (Figure 1). This requires not only efficient catabolic machinery able to generate a large number of reduced coenzymes but also a suitable oxygen supply to receive the electrons travelling through ETC. In the mitochondrial matrix, NADH and FADH_2_ are generated by the tricarboxylic acid (TCA) cycle, also known as the Krebs cycle, and fatty acid β-oxidation reactions. TCA corresponds to a series of chemical reactions involving specific enzymes including several dehydrogenases, this results in the generation of three molecules of NADH and one molecule of FADH_2_ per cycle (Figure 1). The TCA cycle is initiated by citrate synthase (CS) which catalyses the condensation of oxaloacetate, produced after the completion of a round of TCA, and acetyl-CoA, the common end product of all substrate oxidation pathways, including the pyruvate dehydrogenase complex, fatty acid β-oxidation (FAO), the degradation of ketone bodies, and BCAA. As catabolic pathways of the above-mentioned substrates all converge on TCA [7], this process plays a central role in the production of energy molecules in cardiomyocytes as well as in the responsiveness of the energy molecule production machinery, which has to adapt quickly and continuously to heart energy demand. Indeed, while ATP production is paced by its own consumption as ADP stimulates ATP synthase activity [32,33], the regeneration of the reduced coenzymes that support ADP rephosphorylation is accelerated by activation of TCA dehydrogenases. These enzymes, as well as the ATP synthase, are sensitive to the intramitochondrial calcium concentration that increases when cardiac function is challenged since the rate and amplitude of cytosolic calcium transient increases under sympathetic stimulation [34,35,36]. Thus, the perfect coordination between the consumption and production of energy molecules relies on the complementary roles of ADP and calcium which act as signalling molecules to meet energy demand. In addition, the stimulation of TCA reactions by calcium also allows the higher production of compounds leading to NADPH production which is required for reactive oxygen species (ROS) detoxifying systems to avoid oxidative damage that could be really harmful in such an oxidative tissue.

This brief description of cardiac energy metabolism clearly highlights the complexity of the whole process and the multitude of actors, as well as the central role of the mitochondria. It implies fine-tuning to ensure the presence of adequate enzymes and a functional mitochondrial pool over time.

## 3. Energy Metabolism in Healthy Heart

### 3.1. Fatty Acid

FAs are the preferential substrates of the heart to produce energy molecules. Although this source of fuel requires a high number of O_2_ molecules per ATP produced; for instance, the complete oxidation of palmitate generates 105 molecules of ATP and consumes 46 oxygen atoms yielding a P/O ratio of 2.33, allows the generation of large amounts of ATP in conditions where oxygen and mitochondrial capacities are not limiting. This can become an issue in the failing heart, as will be detailed later in this review.

The heart is supplied with blood circulating FAs that must pass several biomembranes to access cardiomyocyte cytosol. Due to their lipophilic property, FAs have long been thought to cross those membranes by passive diffusion. The transfer of these lipids to an intracellular medium is facilitated by membrane-associated fatty acid-binding proteins, usually called fatty acid transporters, such as fatty acid translocase (FAT/CD36) and fatty acid transport protein (FATP) [37]. At the outer mitochondrial membrane, long-chain FAs are transformed into fatty acyl-CoA by long-chain fatty acyl-CoA synthetase and are then transported into the mitochondrial matrix through the carnitine palmitoyltransferase (CPT) system [38]. Short-chain and medium-chain FAs can permeate the mitochondrial membrane in their non-esterified form and are activated to their acyl-CoA derivatives in the matrix by specific mitochondrial acyl-CoA synthetases. There, acyl-CoAs enter the β-oxidation reaction series, producing NADH, FADH_2_, and acetyl-CoA that feeds the TCA cycle for further reducing agent production (Figure 1). The regulation of FA oxidation is quite intricate and occurs at many steps of this catabolic process. While one of the major points of this regulation is the control of FA entry into mitochondria by malonyl-CoA-induced CPT inhibition, the availability and the redox state of FAD/FADH_2_ and NAD/NADH can also play key roles in FA oxidation as they are required for the activity of acyl-CoA transferase and 3-hydroxyacyl-CoA dehydrogenase, respectively [39], two enzymes of the β-oxidation.

### 3.2. Carbohydrates

Unlike FAs, glucose is the most efficient energy substrate from an oxygen utilization point of view, with a P/O ratio of 2.58 (31 ATP produced/12 oxygen atoms consumed). Glucose enters the cardiomyocyte using specific glucose transporters GLUT1 and GLUT4. In the cell, glucose is transformed by hexokinase into glucose-6-phosphate, which initiates diverse series of reactions, including glycolysis (Figure 1), glycogen synthesis, or the pentose phosphate pathway. The production of ATP from glucose starts in the cytosol by metabolizing glucose to pyruvate, which generates 2 ATP molecules. Pyruvate is then either converted to lactate or taken to mitochondria by the mitochondrial pyruvate carrier. In the mitochondrial matrix, pyruvate dehydrogenase (PDH) generates acetyl-CoA using pyruvate, which initiates the TCA cycle, thereby producing reducing agents from carbohydrates (Figure 1). The entry of pyruvate into the TCA cycle is dependent on PDH activity which is accurately regulated by its phosphorylation state; this enzyme is actually activated by PDH phosphatase and inhibited by PDH kinase, which are important players in carbohydrate oxidation regulation [40].

### 3.3. Ketone Bodies

The ketone body (KB) family is an alternative form of energy that is produced by the liver. In normal conditions, the hepatocytes only ensure a moderate production of KBs, which is largely increased when the organism is facing metabolic stress such as a decrease in circulating carbohydrates or/and an increase in circulating FAs. Ketogenesis consists of the production of specific water-soluble lipid molecules called ketone bodies, among which acetone, acetoacetate, and β-hydroxybutyrate are the main species.

The heart preferentially oxidizes β-hydroxybutyrate, the uptake of which is facilitated by monocarboxylate transporter 1 [41]. In cardiomyocytes, β-hydroxybutyrate enters into the mitochondria where it is converted into acetoacetate by β-hydroxybutyrate dehydrogenase; then it is used by CoA transferase succinyl-CoA:3 oxoacid-CoA transferase to produce acetoacetyl-CoA (Figure 1). This latter then undergoes a thiolysis reaction to produce acetyl-CoA, which enters the TCA cycle for further reducing agent production [42]. Although the heart is well-equipped to use KBs, their contribution to energy molecule production is low in normal conditions as KB availability is poor. Yet, β-hydroxybutyrate has a P/O ratio of 2.5, making KBs easier to oxidize than FAs, notably in conditions of reduced mitochondrial oxidative capacities as in HF [7]. This is an important point to consider because KB utilization in the heart is obtained at the expense of FA and glucose oxidation since KB oxidation inhibits catabolic processes involving other kinds of substrates [43,44].

### 3.4. Branched-Chain Amino Acids

Even though the major part of amino acid metabolism occurs in the liver, the subgroup of branched-chain amino acids (BCAA), including leucine, isoleucine, and valine, can be metabolized in non-hepatic tissue such as the heart, brain, or kidney. Briefly, BCAAs enters cardiomyocytes through transporters, which could be specific isoforms of L-type amino acid transporters and bidirectional transporters for L-glutamine and L-leucine/EAA similar to other cells [45,46]. There, they undergo transamination to their respective branched-chain α-keto-acid (BCKA) by branched-chain amino-transferase located in mitochondria. BCKA are then used by branched-chain α-keto-acid dehydrogenase (BCKDH) to produce acetyl-CoA and succinyl-CoA that can respectively enter the TCA cycle and replenish TCA intermediates (anaplerosis) (Figure 1) [47]. The main regulation mechanism involved in the use of BCAA as an energy source is based on the phosphorylation status of BCKDH, which can be phosphorylated by BCKDH kinase and dephosphorylated by mitochondrial-targeted 2C-type ser/thr protein phosphatase (PP2Cm), which controls BCAA-induced dephosphorylation of BCKDH [48]. PP2Cm has been shown to be highly expressed in the heart and is dynamically regulated by stress, thereby demonstrating the importance of BCAAs metabolism in this organ [49].

## 4. Energy Metabolism and B Vitamins

What are the roles of B vitamins in the regulation of energy metabolism? The family of B vitamins is a set of water-soluble vitamins which must be provided by daily diet in a suitable amount to ensure cellular homeostasis. They are involved in a plethora of crucial enzymatic reactions of diverse cellular processes, especially energy metabolism, in which they often act as cofactors or coenzymes for a number of enzymes participating in energy molecule production from the aforementioned substrates [21] (Figure 1). For instance, several B vitamins are directly involved in the TCA cycle, while others are required for NAD/FAD synthesis or for nucleotide synthesis and amino acid metabolism. These vitamins, therefore, play a central role in the energy machinery of the cell, a fortiori in the cardiomyocytes that consume a lot of energy, conferring credit to the hypothesis that vitamin supplementation could be part of the treatment of cardiac diseases associated with major disorders of energy metabolism.

### 4.1. Vitamin B1

Vitamin B1 was one of the first vitamins described following the work of Casimir Funk, who sought to understand the causes of beriberi in the early 20th century. He coined the term vitamin from ‘vita’ (life) and ‘amine’ (nitrogen-containing compound), a nitrogenous substance that is important for life [50]. This vitamin, known as thiamine, is active in its thiamine diphosphate (TDP) form, which is a cofactor of cytosolic transketolase and three mitochondrial enzymes involved in energy production, namely mitochondrial pyruvate dehydrogenase (PDH) [51], α-ketoglutarate dehydrogenase (α-KGDH) [52], and branched-chain α-keto-acid dehydrogenase (BCKDH) [53] (Figure 1). In the brain, a highly oxidative tissue such as the heart, it has been reported that only about 5% of thiamine is bound to cytosolic transketolase while the majority of the vitamin is concentrated in the mitochondria [54], thereby highlighting its major involvement in energy metabolism. Moreover, a recent study reinforced the idea that thiamine is a key element in mitochondrial energy production since thiamine and its derivatives could also allosterically regulate malate dehydrogenase [55], an enzyme involved in the malate aspartate shuttle which transfers reducing equivalents to regenerate NADH in the mitochondrial matrix using NADH produced by glycolysis in cytosol. While the mitochondrial thiamine partners described above clearly make vitamin B1 extremely important for energy homeostasis, its role as a cofactor of cytosolic transketolase is also very interesting since this enzyme catalyzes the first and last step of the pentose phosphate pathway allowing, on the one hand, NADPH production which is required to cellular antioxidant defenses (as a cofactor of glutathione reductase in particular) (Figure 2), and on the other hand, ribose generation for nucleotide synthesis.

In mammals, vitamin B1 cannot be synthesized de novo and is generally found in raw food such as green vegetables, cereals, or egg yolk (a daily intake of 1 mg is recommended). Its intestinal absorption requires a specific transporter (THTR-1), the expression of which is downregulated by alcohol; this explains the fact that alcoholism is one of the major causes of thiamine deficiency in industrialized countries [56,57]. Plasma thiamine is taken up by tissue via THTR1 and is transformed into TDP, the most part of which enters mitochondria via TDP/thiamin antiporter [58], where it binds to the above-mentioned mitochondrial enzymes.

### 4.2. Vitamin B2

Vitamin B2, namely riboflavin, is a major actor of energy metabolism as it is the precursor of flavin adenine dinucleotide (FAD) and flavin mononucleotide (FMN), which act as electron carriers. These two active forms of riboflavin, in particular FAD, are required for the operation of all flavoproteins, which catalyze redox and dehydrogenase reactions that are responsible for many steps in catabolism. Amongst the members of the flavoprotein family, a large part is located in the mitochondria, where they participate more or less directly in the production of energy molecules. This family includes a dozen of mitochondrial acyl-CoA dehydrogenases (ACAD), five of which are short, medium, long and very long-chain acyl-CoA dehydrogenases (SCAD, MCAD, VCAD, VLCAD1, and VLCAD2, also known as ACAD9). These are the enzymes involved in the first step of fatty acid β-oxidation [59,60] (Figure 1). The chemical reactions performed by these ACADs lead to the production of FADH_2,_ the electrons of which are given to ubiquinone (Coenzyme Q_10_ (CoQ_10_)) in the inner mitochondrial membrane through the electron transfer flavoprotein (ETF), thereby feeding the CoQ_10_ pool of mitochondrial ETC in the same way as complex I and II [60]. Riboflavin is also at the heart of the TCA cycle since it supports succinate dehydrogenase (SDH), also known as complex II of the ETC, which catalyzes the two-electron oxidation of succinate to fumarate with the concomitant reduction of CoQ_10_ (Figure 1). During this reaction, FAD bound to the Sdh1 subunit of SDH accepts electrons that are nearly instantaneously channeled through three iron–sulfur clusters of the Sdh2 subunit to ultimately reduce ubiquinone, consequently initiating the transfer of electrons within the ETC [61].

Beyond these essential roles in energy molecule production, riboflavin has a prominent place in the preservation of cellular functions and in particular mitochondrial functions. Indeed, FAD is bound to glutathione reductase (GR), close to the catalytic center [62], where it acts as a temporary acceptor of the electron within GR, which reduces glutathione disulfide (GSSG) to the sulfhydryl form glutathione (GSH) using NADPH (Figure 2). Riboflavin supply is thus required not only for energy metabolism but also for antioxidant defenses of the cell.

This vitamin is found in high quantities in eggs, lean meats, milk, green vegetables, and fatty fish, and the recommended daily intake of riboflavin is approximately 1 mg [63]. In food, riboflavin is found under its FAD and FMN forms which must be hydrolyzed before intestine absorption. This later occurs in the proximal small intestine via specific active riboflavin transporters (RFVTs) and is then released into the plasma to be distributed to the tissues [64].

### 4.3. Vitamin B3

Vitamin B3, the precursor of the NAD coenzyme, was discovered in the early 20th century when deciphering the cause of pellagra, characterized by darkly pigmented skin rash, dermatitis, diarrhea, and dementia [65,66]. Vitamin B3 is found under diverse dietary forms that comprise nicotinic acid (NA) (also known as niacin), nicotinamide (NAM), and nicotinamide riboside (NR) [67]. NAD can also be generated de novo from the essential amino acid tryptophan, essentially in the liver, but it is a less efficient precursor as 60 mg of Trp is considered the equivalent of 1 mg of niacin [67]. The nicotinamide mononucleotide (NMN), a phosphorylated intermediate in the synthesis of NAD can also be considered as a precursor since exogenous NMN administration raises NAD levels in various cell lines and tissues [68], and an NMN transporter (Slc12a8) has recently been identified in the murine small intestine [69]. However, the latter finding has been disputed based on technical consideration and the fact that in many cells, NMN-stimulated synthesis of NAD depends on the action of 5′-ectonucleotidase (also known as CD73) that converts it into NR [70].

NAD is a major coenzyme for fuel oxidation that is involved in diverse steps of anaerobic glycolysis, TCA cycle, β-oxidation, ketones oxidation as well as BCAA degradation (Figure 1). This leads to the donation of two electrons to the oxidized form NAD^+^ producing NADH. In the mitochondria, NADH gives its electrons to mitochondrial complex I, thereby initiating the electron transfer within the ETC, allowing the translocation of the protons to intermembrane space. NAD is, therefore, an essential coenzyme that, together with FAD, constitute electrons carriers allowing the transformation of the energy substrates into usable energy under the form of ATP.

NAD is also the unique precursor of NADP, produced from NAD phosphorylation by NAD kinases in the cytosol (NADK1) and mitochondria (NADK2) [71]. NADP^+^ is reduced to NADPH through different pathways such as the pentose phosphate pathway or the cytosolic and mitochondrial isocitrate dehydrogenases 1 and 2, respectively [72,73] (Figure 1, shown for IDH2 in the mitochondrial matrix). NADPH constitutes the final reducing power for the enzymatic systems detoxifying reactive oxygen species (ROS) (Figure 2). In the mitochondrial matrix, reducing equivalent from the NADH generated from the fuel oxidation process can also be directly transferred to NADP^+^ by the nicotinamide nucleotide transhydrogenase (NNT) to generate NADPH, therefore coupling the production of reducing power for oxidative phosphorylation metabolism (OXPHOS) and antioxidant systems (Figure 1). This is an important mechanism as OXPHOS metabolism is a significant contributor to ROS production [74].

Beyond its role as a major coenzyme in energy molecules production, the oxidized form NAD^+^ also acts as a co-substrate of several enzymes such as sirtuins (SIRT), ADPribose transferases (ART), poly-ADP ribose polymerases (PARPs), and the CD38 ADPribose cyclase that can impact the entire cellular homeostasis. We report readers to a more extensive review on the role of these pathways in the context of heart diseases [75]. Amongst these enzymes, the NAD^+^-dependent deacetylases (sirtuins), in particular, SIRT1 and SIRT3, control the acetylation status of many players of energy metabolism and are consequently major regulators of the cellular energy machinery [76]. For instance, SIRT1 deacetylates and activates the peroxisome-proliferator activated receptor gamma (PPARg) coactivator 1-alpha (PGC-1α), a major transcriptional regulator of metabolic genes and of the mitochondrial biogenesis program [77,78]. As a rise in NAD^+^/NADH ratio stimulates SIRT1 activity, it can be considered as a signaling molecule that participates in the intricate phenomena that coordinate energy demand and production. In contrast with its role as a coenzyme, NAD^+^ is consumed as a substrate by all these signaling pathways (Sirtuins, PARPs, CD38) since it is irreversibly hydrolyzed in NAM and ADP-Ribose and NAD^+^ stores are only replenished by biosynthetic pathways. While homeostatic regulation presides the balance between NAD^+^ hydrolysis and synthesis rates in physiological stages, these two processes sometimes do not match each other in pathological situations, in which a reduction in the steady-state level of NAD^+^ often occurs due to increased activity of consuming enzymes that are not counterbalanced by increased synthesis rate.

In diet, Vitamin B3, mostly in the form of NAM, is found in fish, poultry, and cereals, but a majority of intakes comes from NAD that is hydrolyzed by enzymes at the intestinal brush border to produce NMN, NAM, and NR, as shown in studies on the rat intestine [79,80]. NA derived from gut microbiota-mediated deamidation of NAM could also be an important contributor to NAD homeostasis in the mammalian host [81]. NR enter cells via an equilibrative nucleoside transporter (ENT) family ENT1, ENT2, and ENT4 [82,83]; NA enters via organic anion transporter (OAT)2-mediated transport while NAM entry apparently depends on an unidentified solute carrier (SLC) [84]. As mentioned above, NMN may enter some cells through the SLC12A8 transporter [69] or after being dephosphorylated by CD73 to give NR [85,86].

### 4.4. Vitamin B5

Vitamin B5 (pantothenic acid) is important for energy metabolism since it is the precursor for Coenzyme A (CoA) biosynthesis [87]. As mentioned previously, CoA is required in the catabolism of all types of substrates owing to its function as an acyl group carrier (Figure 1 and Figure 2). CoA is also needed for FA mitochondrial intake through CPT and for KB synthesis in the liver as well as for FA synthesis in the cytoplasm.

Pantothenic acid is found in a wide range of food. It is present in all plant and animal cells. It is found as CoA that is hydrolyzed to pantethine by an intestinal phosphatase before being split to pantothenic acid by an enzyme of the intestinal mucosa. Under this form, vitamin B5 is absorbed by enterocytes through the sodium-dependent multivitamin transporter (SMVT) [88]. Pantothenic acid is then distributed to all organs via the bloodstream since CoA cannot pass through biological membranes. Each cell has to produce its own CoA from pentatonic acid, cysteine, and ATP through a series of five intracellular reactions, or alternatively through degradation of dietary CoA by the ectonucleotide pyrophosphatase (ENPP) [89].

### 4.5. Vitamin B6

Vitamin B6 refers to three distinct molecules (pyridoxal, pyridoxine, and pyridoxamine) leading to the active form of vitamin B6, namely pyridoxal phosphate (PLP). This compound is highly important for the maintenance of mitochondrial functions inasmuch as it is a cofactor of cysteine desulfurase complex which converts L-cysteine to L-alanine and provides the sulfur for iron-sulfur (Fe-S) cluster biosynthesis [90], an essential role of mitochondria that indirectly supports many cellular processes involving Fe-S core proteins (Figure 3). Amongst the latter proteins, mitochondrial ETC complexes contain several Fe-S cores that participate in electron transfer [91,92], thereby placing PLP as a fundamental element for the establishment and maintenance of energy metabolism. This vitamin also acts as a cofactor of many transaminases required for amino acid metabolism [93], such as kynureninase involved in kynurenine-NAD pathways allowing de novo NAD production from tryptophan [94]. Regarding its roles in energy metabolism, PLP takes part in the folate cycle as a cofactor of serine hydroxymethyl transferase, which produces 5,10-methylenetetrahydrofolate [95], an important intermediate in the regeneration of methionine from homocysteine (an amino acid not used in protein synthesis) (Figure 2). Methionine is required for S-adenosyl-methionine, a molecule considered as the major methyl donor supporting protein methylation process, which impacts all cellular pathways, in particular those regulating energy metabolism (see sections “vitamin B9” and “vitamin B12” hereunder for further details) (Figure 2). This specific role of PLP, although well upstream of energy molecule production, makes this vitamin essential for cellular energetics.

In addition to its role in energy metabolism, PLP is involved in a number of pathways. Non-exhaustively, PLP is the cofactor of enzymes involved in cysteine synthesis [96], a proteinogenic amino acid, but also a precursor of glutathione (GSH), a cornerstone of cellular antioxidant systems [97] (Figure 2). The activity of aminolevulinate synthase, a mitochondrial decarboxylase stepping in heme synthesis, also needs PLP [98], placing this vitamin at the heart of the oxygen transport capacity in the blood.

In diet, PLP and its precursors are found in high amounts in vegetables, cereals, and muscle meats. In animal products, these vitamins are mainly present under the forms of PLP and pyridoxamine phosphate, while plant-derived products principally contain pyridoxine and pyridoxine phosphate [99]. These phosphorylated forms of B6 vitamins are hydrolyzed in the intestine prior to their absorption by brush border enterocytes [100]. Although it has long been described that entry of vitamin B6 into intestinal cells was ensured by passive diffusion [100], specific transporters of unphosphorylated B6 vitamines have been suggested since the 2000s [101,102].

### 4.6. Vitamin B7/8

Vitamin B7/8 is known under the name of biotin, which is a prosthetic group of five cellular carboxylases in humans (acetyl-CoA carboxylase 1 and 2 (ACC1, ACC2), propionyl-CoA carboxylase (PCC), 3-methylcrotonyl-CoA carboxylase (MCC), and pyruvate carboxylase (PC)), four of which are located at the outer membrane or in the matrix of mitochondria [103]. Interestingly, all these enzymes are involved in energy metabolism. ACC1 and ACC2 produce malonyl-CoA from acetyl-CoA. While ACC1 is mainly located in the cytosol of lipogenic tissue cells (liver and adipose tissue) and its malonyl-CoA product allows biosynthesis of long-chain fatty acids, ACC2 is highly expressed in the heart and skeletal muscle where it is bound to the outer mitochondrial membrane. There, it modulates mitochondrial long-chain acyl-CoAs uptake [104], as malonyl-CoA is a potent inhibitor of CPT1 preventing FAs influx into mitochondria and subsequent β-oxidation [105]. PCC, MCC, and PC are all located in the mitochondrial matrix. PCC and MCC enzymes regulate critical steps of amino acid catabolism (and odd-chain FAs for PCC) by respectively producing methylmalonyl-CoA from propionyl-CoA for the catabolism of isoleucine, methionine, valine and threonine [106] (Figure 1), and 3-methylglutaconyl-CoA from 3-methylcrotonyl-CoA for leucine and isovaleric acid catabolism [107]. PC also converts pyruvate to oxaloacetate that can be used to replenish the TCA cycle or for the initiation of gluconeogenesis (in liver and kidney) or lipogenesis (adipose tissue, liver as well as brain) [107].

Biotin is found in diet under its active form, which does not undergo further chemical modification within the organism. The main sources of biotin are liver, egg yolk and soybeans; this vitamin is taken up by enterocytes via the same SVMT system as pantothenic acid [88].

### 4.7. Vitamin B9/11

Vitamin B9/11, better known under the name of folate, is involved in a wide variety of chemical reactions. It has originally been identified in the 1930s as an essential nutrient to reduce anemia during pregnancy due to its role in erythropoiesis [108]. Later, it was described as a cofactor of many enzymes participating in various processes such as purine synthesis and methylation reactions [109] (Figure 2). Although folate metabolism is quite intricate, a thorough analysis of the folate cycle reveals clear links between this vitamin and energy metabolism. In mammals, folate acts in the form of many derivatives, which are cofactors that accept or donate one-carbon units (1C metabolism) and are participants in energy molecule production. Amongst these compounds, 5-methyl-THF and 5,10-methyleneTHF play key roles insomuch as they orientate folate metabolism towards specific metabolic pathways (Figure 2). It is important to state that two separate folate cycles coexist in the cell, one in the cytosol and one in the mitochondria with different isoforms of 5,10-methylene-THF dehydrogenases (MTFHR) (Figure 2). In the cytosol, the cycle leads to serine synthesis due to the very high NADPH/NADP^+^ ratio since NADP is the coenzyme of cytosolic MTHFD1. In the mitochondria, the folate cyle leads to serine catabolysis because the mitochondria MTHFD2/L and MTHF1L can use NAD as a coenzyme and the NAD^+^/NADH ratio is high in the mitochondrial matrix. For a detailed review on 1C metabolism, see the review by Ducker and Rabnowitz [110]. The two cycles communicate through the shuttling of reduced THF from the cytosol to mitochondria and the excretion of oxidized formate from mitochondria to cytosol.

The 5-methyl-THF form of the vitamin is used as a methyl-donor to generate methionine from homocysteine in a reaction catalyzed by methionine synthase [111], allowing the regeneration of methionine which is then adenylated to produce S-adenosylmethionine (SAM) (Figure 2). In humans, mitochondrial-derived formate seems to be the primary source of one-carbon units derived from serine catabolysis for remethylation of homocysteine [112]. SAM is considered as the universal donor of methyl groups in methylation reactions consisting in methylation of DNA, RNA, and proteins [113]. More than a hundred cellular compounds can be subjected to this methylation process, including important regulators of energy metabolism such as PGC-1α or SIRT1. Indeed, PGC-1α can be methylated and activated by protein arginine methyltransferase 1 (PRMT1) [114], while the expression and/or activity of SIRT1 could be directly or indirectly impacted by these transmethylation reactions [115]. This clearly confers an important role to folate in cellular energetics. It is noteworthy that methionine is also required for GSH synthesis, and, in this sense, folate is an important element of the cell’s antioxidant defenses especially as this vitamin acts as a direct antioxidant and scavenger molecule [116]. The 5,10-methyleneTHF form, for its part, is used for thymidylate and purine synthesis [109]. For this reason, folate is not only essential for cellular proliferation but also for ATP production that requires prior synthesis of adenine, a purine nucleobase. In addition, 10-formyl-THF is used for the formylation of mitochondrial initiator methionine tRNAs that is required for translation of mitochondrially encoded proteins [117].

Folate is abundant in green vegetables, oranges, eggs, or unprocessed grains (a daily intake of 0.4 mg is recommended). It is mostly present in the reduced form in the human diet, typically 5-methyl-THF [118]. In diet, it is mostly bound to proteins as polyglutamates that must be hydrolyzed by proteases before being absorbed in the small intestine. Folate uptake into enterocytes of brush-border requires the presence of specific transporters such as proton-coupled folate transporter (PCFT), the mutation of which has been identified as a cause of hereditary folate malabsorption [119].

### 4.8. Vitamin B12

Vitamin B12, namely cobalamin (Cbl), exhibits a quite complex structure when compared with the others B vitamins. It is required as a cofactor of cytosolic and mitochondrial enzymes that are directly or indirectly involved in energy metabolism. In the cell, Cbl is transformed into methylcobalamin and adenosylcobalamin, which are respectively found in the cytosol and in the mitochondria [120]. Methylcobalamin acts as a cofactor in the reaction catalyzed by methionine synthase in which it plays, similar to 5-methylTHF, the role of methyl group carrier to support the conversion of homocysteine to methionine [113] (Figure 2). Owing to this role in methionine synthesis, cobalamin is at the origin of the methylation process and GSH production in a similar way as folate. Adenosylcobalamin, under the form of deoxyadenosylcobalamin, is a cofactor of methylmalonyl-CoA mutase, which converts methylmalonyl-CoA, derived from carboxylation of propionyl-CoA, to succinyl-CoA within the mitochondrial matrix [121] (Figure 1). This pathway is important for the degradation of amino acids (valine, isoleucine, and methionine) and metabolites of odd-chain fatty acid, allowing replenishment of the TCA cycle from these sources.

The main sources of Cbl are organ meats, milk (and milk products), and shellfish so that strict vegetarians need to take Cbl as a supplement, although the daily recommended intake is a few µg. The highly acid gastric environment plays a key role in Cbl absorption as acidity and pepsin allow Cbl release contained in food. Cbl then binds to intrinsic factor (IF) produced and secreted by acid-secreting parietal cells of the stomach [122]. The IF-Cbl complex is absorbed in the distal ileum via the receptor for IF-Cbl, cubilin [123].

## 5. Alterations of Energy Metabolism in Heart Failure

Drastic alterations of energy metabolism have largely been reported in the pathophysiology of HF and are now thought to be a key element of the progression of the disease. Although metabolic phenotype in HF can perceptibly vary according to etiology and HF stage as alterations progressively set in [124], studies in human and animal HF models revealed typical metabolic alterations in advanced HF. Amongst these alterations, the metabolic switch from FA utilization to carbohydrate metabolism, which consumes less oxygen to produce ATP, has extensively been described in the last decades [125,126].

While FA oxidation would not be systematically affected at the early stage of HF (depending on the model) [127,128], advanced and end-stage HF are clearly characterized by a strong decrease in FA utilization, especially due to the down-regulation in FA oxidation enzyme expression (LDAC and MCAD especially) shown in animal HF models and human [129,130]. The mechanisms at the origin of these dysregulations in HF are still not completely understood, but they undoubtedly involve a reduction in the protein level of PPARα [131] and its transcriptional co-activator PGC-1α [132], as PPARα plays a pivotal role in regulating the expression of genes encoding actors of mitochondrial β-oxidation and FA transport [133]. This alteration of FA metabolism regulators is a part of the so-called fetal reprogramming that is a well-known hallmark of HF [134] and is characterized by a switch of substrate preference of cardiomyocytes to carbohydrates, as is the case in the immature heart [135]. The increase in carbohydrate utilization as an alternative source of ATP production seems to be an earlier event than the decrease in FA oxidation in the progression of cardiac diseases to HF since studies report higher capacities to use glucose as soon as compensated cardiac hypertrophy stage [136,137]. The underlying mechanisms initiating the stimulation of carbohydrate catabolism processes are still not perfectly understood, although AMPK, the energy stress sensor kinase, could play a determinant role in this phenomenon [138].

The utilization of glucose is beneficial and improves the efficiency of the heart as long as cardiac oxidative capacities allow glucose oxidation [139]. However, as HF evolves, alterations of mitochondrial functions [3] and modulations of pyruvate metabolism (probably due to changes in PDH activity/expression, pyruvate transport and NADH shuttle [140]) occur and lead to a stage in which this switch is no longer adapted to support suitable energy production. Finally, the increase in the capacities of the cardiomyocytes to produce energy from glucose is only transitory [128]; the worsening of HF is actually associated with a decrease in energy production from glucose owing to major dysregulations such as alterations of mitochondrial machinery, i.e., TCA cycle and oxidative phosphorylations [141]. This results in the loss of the coupling between glycolysis and glucose oxidation which is required for efficient use of this substrate [9]. A general decrease in mitochondrial mass and mitochondrial biogenesis also contributes to the decreased oxidative capacity in the failing heart, partly due to the decrease in PGC-1α and its transcription cascade [132]. Interestingly, glycolytic enzymes expression has also been reported to be decreased in an animal at the advanced stage of HF [142], suggesting that global alteration of glucose catabolic processes takes part in the collapse of cardiomyocyte energy metabolism and in the energetic distress of the failing myocardium. As a result, at the end-stage, the failing heart contains about 30% less ATP than a healthy heart and also exhibits a lower PCr content [143], PCr being generated by creatine kinase from ATP and acting as an energy storage molecule and an energy transport compound (for further details see [144]).

The defects in mitochondrial functions, in particular in oxidative capacities, are major causes of energy deficit in HF. These alterations are the consequences of combined phenomena such as compromised mitochondrial biogenesis and increased reactive oxygen species (ROS) production by mitochondrial electron transfer chain. HF is indeed associated with profound perturbations of mitochondrial life cycle partly due to down-regulation of PGC-1α and its downstream target genes, which control the renewal of the mitochondrial pool of cardiomyocytes [132], thereby leading to the accumulation of mitochondrial defects reported in the failing myocardium. As the quality of the mitochondria decreases, these organelles produce more and more ROS, such as superoxide and hydrogen peroxide, that alter the structural component of the cardiac cell and precipitate the cellular damage [145].

## 6. B Vitamins in Heart Failure

It is now well accepted that patients with HF present micronutrient deficiencies owing to malabsorption resulting from splanchnic congestion, increased urinary loss of nutrients caused by diuretics largely used in HF treatment, as well as a poor oral intake at advanced stages of HF [146,147,148]. These last decades, the micronutrient status of HF patients has been the subject of many studies, and it has been shown in particular that B vitamins deficiency has been found to be more prevalent in HF patients than in the general population [23,149]. This has been described almost for the whole family of B vitamins, and one can reasonably think that a deficiency in one or more of these vitamins could participate in the deficit of energy production reported in HF. This is why many research teams have suggested that the addition of supplementation with specific B vitamins to standard HF therapies could be beneficial and could potentially help to preserve the cardiac function of HF patients longer. As several B vitamins are at the heart of many energy metabolism processes, it is thus conceivable that the use of certain vitamins could be beneficial to support the activity of the failing heart even in patients with no specific deficiency as these vitamins could play a role in boosting particular metabolic pathways to restore the energy balance of the myocardium. Beyond their potential impacts on energy metabolism, the use of B vitamins could also be interesting in the context of cardiovascular disease owing to their effects on the vascular system and atherosclerosis [150].

### 6.1. Vitamin B1

Thiamine deficiency in HF has been regularly reported [14,151,152] and could reach about 90% of hospitalized HF patients receiving furosemide [153,154], a loop diuretic which is thought to participate in this deficiency [155,156], although this hypothesis has not been systematically confirmed in trials [151]. It is difficult to accurately assess the rate of thiamine deficiency in HF as the available data comes from studies with very specific designs, including different HF populations (age range, medications, underlying nutrition status…). The fact that a significant proportion of the general HF population is affected by this deficency, and that mitochondrial energy production function is intimely related to thiamine, gives weight to the hypothesis that this vitamin could be useful in HF therapy.

The effects of thiamine or benfotiamine (a thiamine prodrug with high bioavailability) on the progression of cardiovascular diseases have been largely studied for twenty years. Using animal models, it has been clearly shown that these compounds protect the function of the heart facing stress such as ischemic injury [157], myocardial infarction [158,159], or doxorubicin cardiotoxicity [160]. It has also been suggested that vitamin B1 would be particularly efficient to prevent diabetes-induced diastolic dysfunction in type 1 and type 2 diabetic mice [161]. Interestingly, the benefit of vitamin B1 supplementation on cardiac function would especially rely on the reduction of oxidative stress [158,159,162] as well as the preservation of mitochondrial functions [157,163]. As mentioned above, mitochondrial defects and dysregulation of ROS production/detoxification are major alterations in the failing myocardium; it thus confers credit to the use of thiamine and/or its derivatives to treat HF. Incidentally, the benefits of thiamine supplementation in HF treatment have been assessed in small groups of patients. Several trials, including a maximum of a few tens of HF patients, supplemented with thiamine (dose ranging from 100 mg to 300 mg per day), reported a significant increase in left ventricular ejection fraction after several weeks of supplementation [153,164,165]. Whereas the trial led by Seligmann et al. in the nineties showed that the beneficial effect of thiamine on left ventricular ejection fraction was associated with better functional capacity (improved by at least one NYHA class) [153], it was confirmed in a more recent study by Schoenenberger et al. in which no change in walking time was observed in patients with improved left ventricular ejection fraction after thiamine supplementation period [165]. This raises questions about the real usefulness of thiamine supplementation in HF therapy, especially as other trials failed to demonstrate any benefit of vitamin B1 (dose ranging from 100 mg to 300 mg per day) although thiamine circulating level increased [15,166,167,168]. These studies not only failed to show a beneficial effect on cardiac function (FE) [15,167] but also failed to show an improvement in the quality of life of HF patients [15,166,167,168]. So far, the evaluation of thiamine supplementation as a new tool in HF therapeutics has shown mixed results and conclusions from the aforementioned studies are not clear because of the small sample size, short duration of follow-up, lack of any dietary assessment of thiamine intake, as well as the lack of precisely defined inclusion criteria. This does not lead to the recommandation for the routine use of thiamine therapy in general HF population but larger trials should be led to clearly state.

### 6.2. Vitamin B2

The rare assessments of riboflavin status in patients with HF suggest that the prevalence of vitamin B2 deficiency would be higher in this population [23,146,149], especially due to lower intake [169,170]. However, to our knowledge, no trial was specifically designed to study the effects of riboflavin supplementation in HF patients or even to understand the potential links between this deficiency and cardiac dysfunction. In elderly HF patients, this vitamin has actually been tested as part of micronutrient supplementation, including fifteen compounds (vitamins and minerals) [171]. While this cocktail significantly improved the left ventricular function of the heart in those patients, it would be speculative to suggest a beneficial effect of riboflavin at this point.

The most convincing results suggesting that riboflavin may be beneficial/protective for cardiac function have been obtained in animal models. Riboflavin has been shown to alleviate myocardial hypoxic/ischemic injury in mice via the activation of lysine-specific demethylase 1 (LSD1), which requires FAD to ensure its epigenetic modification function [172]. In this study, the authors reported that riboflavin supplementation increased FAD production/levels in vivo. Given the importance of flavoprotein in energy metabolism, the administration of riboflavin for its role as a FAD precursor could be useful to support energy production in situations of energy distress such as HF. Although the physiological importance of FAD, in particular in cardiac metabolism, has been overlooked in comparison with NAD, FAD could yet be of great interest to restore energy metabolism. For instance, it has been demonstrated that FAD stabilizes mitochondrial acyl-CoA dehydrogenases (SCAD and MCAD) activity in vitro [173]. In spontaneously hypertensive rats, treatment with FAD inhibited pathological cardiac hypertrophy and fibrosis and these effects were associated with a significant increase in SCAD activity, higher ATP content, and a decrease in ROS level [174], thereby suggesting a cardiac protective effect of this compound that should be considered at least to hinder the progression of cardiovascular diseases to HF. Riboflavin has also proven its abilities to protect cardiac function in a rat model of type 1 diabetes induced by streptozotocin in which the oral administration of riboflavin preserved myocardial function and improved heart antioxidant status [175].

To date, there is a clear lack of strong evidence to determine if riboflavin could be useful to treat HF patients, but the limited data available in the literature suggest a cardioprotective role in animals and must prompt further investigations.

### 6.3. Vitamin B3

In the last decade, a number of studies in rodent models of HF including myocardial infarction (MI) triggered by experimental ligature of the left coronary artery, pressure-overload hypertrophy triggered by experimental transverse aorta constriction (TAC), or genetic models of cardiomyopathy showed that NAD homeostasis is altered in the failing heart. The findings either showed a decrease in the NAD^+^/NADH ratio in the context of TAC or complex I deficiency due to accumulation of NADH [176,177] or a decrease in the global pool of NAD (i.e., the sum of NAD^+^ and NADH pools) [178,179,180,181]. Even in recently developed models of HF with preserved ejection fraction (HFpEF), NAD homeostasis appears to be altered [182,183]. At least one study reported a similar 30% decrease in NAD levels in failing human hearts from DCM patients [184]. The cause for the drop in the myocardial NAD level is not clearly understood and could vary depending on the model. It is potentially due to more active NAD^+^-signaling pathways that cleave the NAD molecules and that are activated by energy and oxidative stress (e.g., SIRT1, PARP1). However, cardiac expression of the nicotinamide phosphoribosyl transferase (NAMPT) is also almost systematically repressed in HF models and could lead to deficient NAD biosynthesis [185,186]. Interestingly, the alternative NAD synthetic pathway mediated by nicotinamide riboside kinase 2 (NMRK2) appears to be upregulated to compensate for the drop of NAMPT expression, especially in mouse DCM models [179,186]. This striated muscle-specific kinase has been described to be essential to maintain NAD levels in the murine aged heart [187].

In all the studies cited above, supplementation of conventional rodent diet with therapeutic doses of diverse forms of vitamins B3 (NR, NAM, or NMN ranging from 200 to 400 mg/kg of body weight/day) resulted in increased myocardial NAD levels, beneficial outcome in terms of survival and improvement of ejection fraction in models of HF with reduced ejection fraction (HFrEF) or diastolic parameters in HFpEF. In most but not all of the preclinical studies, boosting NAD levels were also associated with improved mitochondrial oxidative capacities in the heart. Other mechanisms linked to the NAD^+^-dependent signaling pathways probably play important roles in these beneficial effects, and the precise molecular mechanisms remain to be better explained.

Altogether these positive effects have been raising hope for bench-to-bed transfer and clinical applications [75,188,189]. A few pilot phase 1/2 trials assessing the effect of NR supplementation (2000 mg/day) on systolic HF patients have been completed (e.g., NCT03423342, First Posted: 6 February 2018 and NCT03727646, First Posted: 1 November 2018 at https://clinicaltrials.gov), but results regarding the HFs symptoms were not published yet at the time of this review. Of note, NR administration was reported to suppress inflammatory activation of peripheral blood mononuclear cells (PBMCs) in HF, thereby suggesting that beyond targetting NAD deficiency in the failing heart, NR treatment may have interesting systemic effects [190].

### 6.4. Vitamin B5

Despite the importance of pantothenic acid in acetyl-CoA biosynthesis and consequently in metabolism, this vitamin is not routinely measured in patients, so that there is no data on the prevalence of pantothenic acid deficiency in HF patients. Incidentally, vitamin B5 supplementation in the context of heart disease has apparently not been investigated largely even in multi-micronutrient supplementation studies. Yet, during the 1990s, a young boy affected by a dilated cardiomyopathy due to type 2 X-linked 3-methylglutaconic aciduria was treated with pantothenic acid that showed impressive positive effects on heart function [191], demonstrating the therapeutic potential of this compound in particular conditions.

A handful of studies looked at the effect of this vitamin on cardiac function in animal models inducing cardiac stress. In rats, dexpanthenol, a precursor of pantothenic acid, is beneficial for the heart during sepsis induced by cecal ligation and puncture [192] and could also protect the heart from isoproterenol-induced cardiac damage [193]. This protection could rely on the preservation of antioxidant machinery that would maintain a better antioxidant status of the myocardium; a mechanism already suggested to explain the protective effect of dexpanthenol on cardiovascular damage, especially on endothelial dysfunction, in the type 1 diabetes rat model [194]. Seemingly, vitamin B5 has not really been considered as a potential stimulator of energy metabolism in the context of diseases in which major alterations of energy production have been described; this question remains open for further research.

### 6.5. Vitamin B7/8

To the best of our knowledge, HF has not been associated with a high prevalence of biotin deficiency, and no study has investigated the potential benefit of biotin supplementation in cardiac diseases. The understanding of the role of biotin in various disease states is quite limited [23]. Although cardiac tissue seemed to be relatively insensitive to biotin deficiency in rats [195], the role of this vitamin as a prosthetic group of several key enzymes of energy metabolism suggests that this compound and its role in HF pathophysiology deserves more attention and should be the subject of further investigations.

### 6.6. Vitamin B6, Vitamin B9 and Vitamin B12

Pyridoxine, folate and cobalamin have largely been studied for their roles in homocysteine metabolism, and, for this reason, they have often been used together to test their benefits in the treatments of pathologies associated with hyperhomocysteinemia (HHCY) [196,197,198]. HHCY has been reported in HF regardless of its etiology [199,200], and circulating homocysteine level is related to clinical variables of HF [201,202], thereby suggesting a link between this compound and the clinical status of the patients. HHCY is generally caused by a deficiency in vitamin B6, B9, and/or B12 [198,203,204]. Although HHCY has clearly been described in HF, deficiency in these vitamins are under question. It seems that pyridoxine level would be decreased in HF patients [149,205], but folate and cobalamin deficiency would be relatively rare [206], and these vitamins would not be related to the severity of the disease [201]. However, a recent study reported a subclinical cobalamin deficiency associated with increased serum methylmalonic acid in HF and suggests that vitamin B12 deficiency might have been underestimated in previous clinical studies [203]; these data should be confirmed by larger studies. HHCY in HF could also be a consequence of kidney dysfunction associated with HF [207] since this organ is responsible for homocysteine clearance. Reduced urinary homocysteine excretion can be a cause of homocysteinemia elevation when folate and cobalamin levels are normal [208]. Whatever the cause of HHCY in HF, it could be part of the mechanisms leading to the alteration of the general condition of HF patients. Deleterious effects of HHCY on cardiac function could be indirect as a high level of homocysteine is known to affect the vascular system. For instance, it induces endothelial cell dysfunction and modulates vascular smooth muscle cells contractility. This can especially be explained by the fact that HHCY leads to the inhibition of nitric oxide synthesis [209], increases ROS production [210] and stimulates matrix metalloproteinase activity [211] in endothelial cells as well as enhances collagen production by smooth muscle cells [212]. These dysfunctions probably participate in the well-known development of atherosclerosis in patients with high homocysteinemia that presents HHCY as an independent risk factor for atherosclerosis [213]. In addition to its atherothrombotic effect, HHCY could also directly affect cardiac function. In animals, diet-induced HHCY has been shown to induce important myocardial collagen deposition associated with diastolic dysfunction in hypertensive rats [214] or ventricular (left and right) hypertrophy with increased perivascular and interstitial collagen as well as myocardial mast cell infiltration in normotensive rats [215]. It has even been demonstrated using coronary-perfused hearts that homocysteine elicited a concentration-dependent negative inotropic action that was partially antagonized by a non-selective antagonist of adenosine receptor [216,217].

Owing to HHCY effects on vessels and the evidence suggesting that HHCY contributes to the development of cardiovascular diseases, homocysteine-lowering interventions have been proposed to prevent cardiovascular events, especially stroke and myocardial infarction. Thus, many studies consisted in giving vitamin B6, B9, or B12 (alone or in combination) to adults at risk of or with established cardiovascular disease these last two decades. Generally, these studies and post hoc meta-analysis failed to demonstrate clear beneficial effects of these vitamins on cardiovascular events whatever the vitamin combination, although the administration of pyridoxine, folate, and/or cobalamin often significantly lowered homocysteine levels [113,197,218,219]. On the other hand, a study led by Towfighi et al. showed that, after stratifying the treated population by age, vitamin supplementation was associated with reduced risk of stroke and myocardial infarction in patients older than 69 [220], suggesting that older patients are more likely to benefit from this B vitamin therapy. Another meta-analysis including studies testing supplementation with folate alone demonstrated that it slightly reduced the risk of stroke and CVD, with more reduction for the risk of CVD in patients without preexisting CVD or with a large decrease in homocysteine levels [221]. This beneficial effect of folate on the risk of stroke has been confirmed by a recent meta-analysis that, however, mentioned that folate supplementation did not reduce the risk of coronary heart disease [222]. Whereas these conflicting results did not lead to clear recommendations to use these vitamins in patients, it seems that even if the effect of these three vitamins on the prevention of cardiovascular events is not fully established; they could be of great value to treat patients with specific profiles that have to be determined by further studies.

To our knowledge, this homocysteine-lowering strategy has not been tested in patients with established HF. Given the prevalence of HHCY in HF and the deleterious effects of HHCY on cardiac function, it is surprising that this kind of study has never been considered and that data related to the effects of these vitamins in HF are lacking. Besides, the use of these vitamins in animal models of cardiac dysfunction highlighted other actions of these compounds, suggesting that they could be useful to preserve heart function independently of their homocysteine-lowering effect. For instance, folate has been shown to protect cardiac function in mice subjected to myocardial infarction [223], doxorubicin [224], celecoxib [225], high fat diet-induced obesity [226], and would also prevent age-related cardiac remodeling [227]. It could act through the normalization of metalloproteinase inhibitor levels [223], modulation of endothelial nitric oxide synthase [224], or the reduction of cardiac oxidative stress [226]. Its beneficial effects in old mice would especially be related to the modulation of ER stress pathway, which is usually upregulated in the aged heart [227]. Folates would protect diastolic function and decrease overall myocardial collagen deposition in a rat model of hypertrophy induced by monocrotaline [228]; its antioxidant properties could play an important role in the protective effect of folate in this model [229]. Cobalamine has been identified among activators of the PGC1α, a regulator of mitochondrial biogenesis and genes related to FAO and glycolysis, as mentioned previously [230]. A combination of folate and cobalamin has also been successfully used in rodents, in which it reduced cardiac damage in a model of isoproterenol-induced myocardial infarction in rats with HHCY [231], proving the ability of such supplementation to lower homocysteine levels with subsequent protection of the heart. A recent study testing the effects of folate and cobalamin supplementation in a mouse model of heart failure induced by pressure overload showed that left ventricle ejection fraction was preserved in mice fed with food supplemented with folate and cobalamin [232]. Interestingly, this was associated with a preservation of mitochondrial function that could be due to a minimized alteration of the mitochondrial biogenesis process. The underlying mechanisms would involve the role of folate and cobalamin in S-adenosylmethionine production which could then support methylation reactions that would stimulate PGC-1α activity in this model. Beyond the fact that this study showed that a supplementation with folate and cobalamin was beneficial for the heart facing pressure overload, the preservation of energy metabolism described in this work suggests that B vitamins could be part of a metabolic therapy of HF. Nowadays, the use of these vitamins in HF patients is not really considered because of the lack of convincing data in human. However, the improvement of endothelial function in cardiac transplant recipients and HF patients receiving pyridoxine and folate supplementation, respectively, demonstrate the benefit of these vitamins on the cardiovascular system in pathology [233,234].

All the above-mentioned preclinical and clinical studies and the main effects of B vitamins that could be beneficial in the context of heart failure are summarized in Table 1 and Figure 3, respectively.

## 7. Conclusions

Alterations of cardiac energy metabolism are clue factors in the etiology of heart failure. We have presented ample evidence that B vitamins are important factors involved at all levels of the regulation of cardiac energy metabolism. This review of the literature shows that B vitamins potentially present great interest in the regulation of cardiac energy metabolism and thus treatment of cardiovascular diseases. Studies in animals have shown that these vitamins have beneficial effects on cardiac energy metabolism and thus cardiac function. Today their use in humans in the context of heart failure is hampered by the inconclusive results of low-power clinical studies, whose design was more often oriented toward prophylactic measures of risk reduction rather than a therapeutic approach despite studies showing an improvement of cardiac function and/or the quality of life of the patients. However, certain vitamins were never even tested within the precise context of HF. In view of the positive results mentioned above and their importance in cardiac energy metabolism, it seems that vitamins could be useful in patients whose profile remains to be defined. Advanced age and malnutrition will be important but not the sole criteria for their indication. Boosting effects on vitamin-dependent signaling pathways could be of interest for patients outside this category. To implement this approach in the near future, it is important to better understand the benefits of these vitamins at the molecular/cellular level by increasing the number of studies in animal models of HF and then conducting clinical trials in HF patients rigorously selected according to criteria established on the basis of the knowledge emerging from those preclinical studies. The integration of B vitamins in the therapeutic arsenal of HF should therefore be considered seriously; these vitamins likely have a strong therapeutic potential that will only be fully exploited when their use is based on better knowledge.

## Figures and Tables

**Figure 1 ijms-23-00030-f001:**
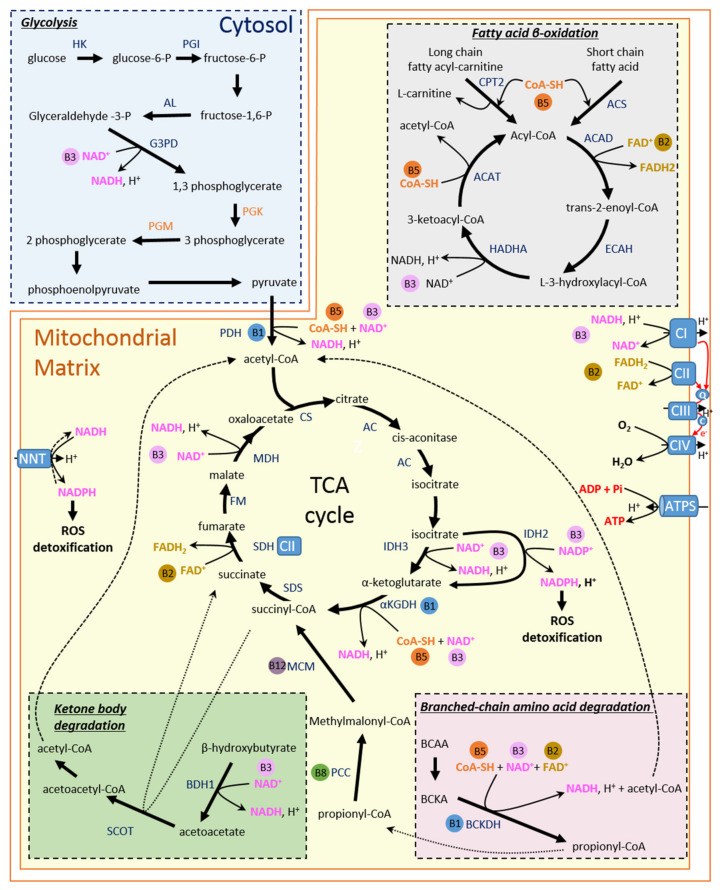
Role of B vitamins in energy metabolism processes in cardiomyocytes. AC, aconitase ACAD, acyl-CoA dehydrogenase; ACAT, acyl-CoA thiolase; ACS, acyl CoA synthetase; AL, aldolase; BCAA, branched-chain amino acid; BCAT, branched-chain amino acid transaminase; BCKA, branched-chain keto acids; BCKDH, branched-chain keto acid dehydrogenase; BDH1, beta-hydroxybutyrate dehydrogenase; CPT2, carnitine O-palmitoyltransferase 2; CoA-SH, Coenzyme A; CS, citrate synthase; ECAH, Enoyl-CoA hydratase; EL, enolase; FAD^=^, flavin adenine nucleotide; FADH_2_, reduced form of FAD^+^; G3PD, glyceraldehyde-3-phosphate dehydrogenase; FM, fumarase, HADHA, L-3-hydroxyacyl-CoA dehydrogenase; HK, hexokinase; IDH, isocitrate dehydrogenase; αKGDH, alpha-ketoglutarate dehydrogenase; MCM, methylmalonyl-CoA mutase; MDH, malate dehydrogenase, NAD^+^, nicotinamide adenine dinucleotide; NADH, reduced form of NAD^+^; PCC, propionyl-CoA carboxylase; PDH, pyruvate dehydrogenase; PGI, phosphoglucose isomerase; PGK, phosphoglycerate kinase; PGM, phosphoglycerate mutase; PK, pyruvate kinase; SCOT, succinyl-CoA:3 oxoacid-CoA transferase SDH, succinate dehydrogenase; SDS, succinyl-CoA synthase; TCA, tricarboxylic acid.

**Figure 2 ijms-23-00030-f002:**
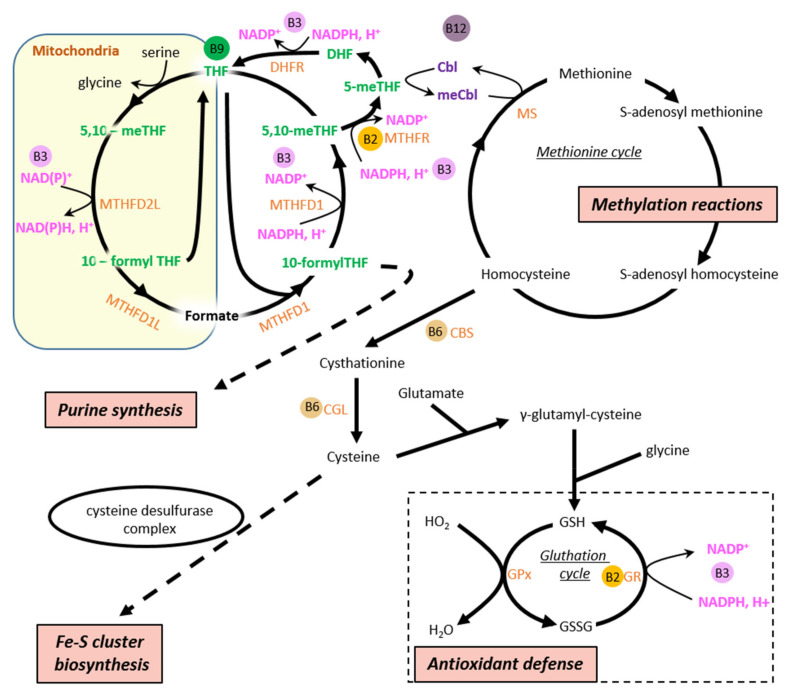
Involvement of B vitamins in processes impacting energy metabolism. CBS, Cysthationine β synthase; CGL, Cysthationine γ-liase; GPx, glutathione peroxidase; GR, glutathione reductase; GSH, glutathione; GSSG, glutathione disulfide; MS, methionine synthase; 5-MTHF, 5′ methymtetrafolate; 5,10-MTHF, 5,10-methylenetetrafolate; THF, tetrafolate; NADP^+^, nicotinamide adenine dinucleotide phosphate; NADPH reduced form of NADP^+^.

**Figure 3 ijms-23-00030-f003:**
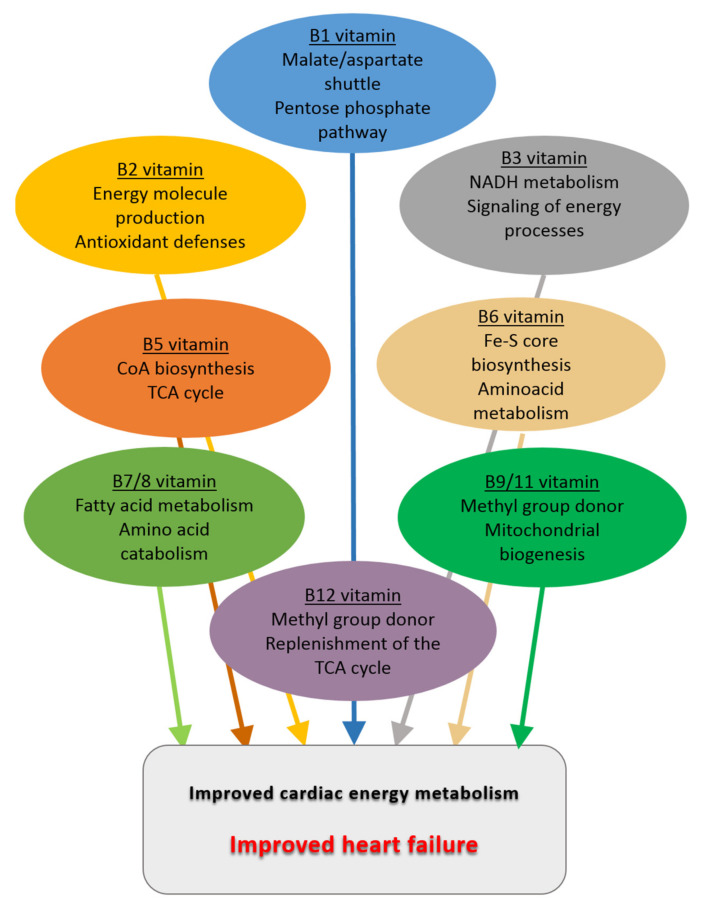
Main effects of B vitamins on energy metabolism and their potential benefits in heart failure.

**Table 1 ijms-23-00030-t001:** B vitamin supplementation and cardiovascular diseases in animals and humans.

Vitamin	Specie	Effects	Studies
B1	Mice	Positive effects on cardiac function:	
		- myocardial infarction	[158]
		- diabetes-induced cardiac dysfunction	[161]
	Rat	Positive effects on cardiac function:	
		- ischemic injury	[157]
		- myocardial infarction	[159]
		- doxorubicin cardiotoxicity	[160]
		- diabetes-induced cardiac dysfunction	[163]
	Human	Supplementation (100 to 300 mg/day) in HF patients:	
		- increase in left ventricular ejection fraction	[153,164,165]
		- better functional capacity (NYHA class)	[153]
		- no change in walking time	[165]
		- no benefit on cardiac function (FE)	[15,167]
		- no improvement in the quality of life	[15,166,167,168]
B2	Mice	Supplementation with riboflavin: reduction of myocardial ischemic injury	[172]
	Rat	FAD treatment decreases cardiac hypertrophy and fibrosis in SHR rats	[174]
		Supplementation with riboflavin: protect heart function (type1 diabetes)	[175]
	Human	Supplementation with a cocktail of vitamins and minerals, including riboflavin: improvement of ventricular function	[171]
B3	Mice	Supplementation with NMN	
		- preserves of cardiac mitochondrial function in complex-I deficient mice exhibiting accelerated HF in response to chronic stress	[176]
		- delays the development of HF in mice with mitochondrial dysfunction	[177]
		Supplementation with NR	
		- preserves cardiac function in *Srf* mutation induced-DCM	[179]
		- preserves cardiac function in *Lmna* mutation induced-DCM	[180]
		- improves cardiac mitochondrial function and ameliorates HFpEF phenotype	[183]
		Supplementation with nicotinamide improves diastolic dysfunction induced by aging	[182]
		Exogenous NAD blocks cardiac hypertrophy response	[235]
B5	Rat	Supplementation with nicotinamide improves diastolic dysfunction induced by hypertension or cardiometabolic syndrome	[182]
	Human	oral NR administration: improvement of PBMC respiration and reduced proinflammatory cytokine gene expression in patients with HF	[190]
	Rat	Treatment with dexpanthenol, a vitamin B5 precursor:	
		- protection of the heart during sepsis	[192]
		- protection heart from isoproterenol-induced damage	[193]
		- beneficial effects on endothelial function (type 1 diabetes)	[194]
B6/B9/B12	Mice	Folate supplementation protects cardiac function:	
		- myocardial infarction	[223]
		- doxorubicin cardiotoxicity	[224]
		- high-fat diet-induced obesity	[226]
		Supplementation with folate and cobalamin: Preservation of left ventricular ejection fraction (pressure overload induced-HF)	[232]
	Rat	Folate supplementation protects cardiac function: celecoxib cardiotoxicity	[225]
		Folate supplementation protects diastolic function and prevents fibrosis: monocrotaline-induced hypertrophy	[228,229]
		Supplementation with folate and cobalamin: reduction of cardiac damage (isoproterenol-induced infarction)	[231]
	Human	Supplementation with B6/B9/B12: decrease in risk of stroke and myocardial infarction in patients older than 69	[220]
		Supplementation with folate:	
		- decrease in risk of stroke	[221,222]
		- decrease in risk of CVD in patients without preexisting CVD	[222]
		Pyridoxine: improvement of endothelial function (cardiac transplant recipients)	[233]
		Folate: improvement of endothelial function (HF patients)	[234]

## Data Availability

Not applicable.

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
