# Peer review of "Metabolic Therapy of Heart Failure: Is There a Future for B Vitamins?"

_ijms, 2021, doi:10.3390/ijms23010030_

Round 1

Reviewer 1 Report

The work, entitled " Metabolic therapy of heart failure: Is there a future for B vitamins?," aims to aims at assessing the value of B vitamin supplementation in the treatment of HF. A wide range of topics are covered, all of which are relevant ", highlights the importance of vitamin B. The paper is well written and has the merit of publication. However, some issues require attention:

-A resume integrative picture could be done to provide an illustration of data and transpose it.

-How can be translated to clinical setting?

Author Response

Comments and Suggestions for Authors

The work, entitled " Metabolic therapy of heart failure: Is there a future for B vitamins?," aims at assessing the value of B vitamin supplementation in the treatment of HF. A wide range of topics are covered, all of which are relevant ", highlights the importance of vitamin B. The paper is well written and has the merit of publication. However, some issues require attention:

A resume integrative picture could be done to provide an illustration of data and transpose it.

Response: As suggested by the referee we have added a new figure (Figure3) which summarizes the role of B vitamins in energy metabolism and their potential therapeutic effect to restore cardiac energy metabolism and improve heart failure.

How can be translated to clinical setting?

Response: This is presented in the Conclusions and summarized in Figure 3.

Reviewer 2 Report

This review written by Piquereau and colleagues entitled “Metabolic therapy of heart failure: Is there a future for B vitamins?” describes various aspects of heart failure with a focus on energy metabolism. The authors highlight the origins and known molecular functions of all known B vitamins in this lengthy review. Consequently, the authors discuss how each B vitamin might improve cardiac function during heart failure. Ultimately, the authors conclude that clinical studies have not been able to indicate a beneficial effect of B vitamins to date. The manuscript pertains to in-depth discussions of B vitamins and their potentially beneficial function in the cardiovascular system. The author convincingly demonstrate their expertise on the topic and make a strong case for further studies on B vitamins as a treatment modality. However, some points should be addressed to improve the manuscript.

  1. A principal point of criticism is regarding the novelty of the review in general. Most of the studies described in the review have been discussed extensively in other reviews and the vast majority of the references have not been published in the past 5 years. To illustrate this point: only about 30% of the references were published in 2016 or later (excluding the introduction, which mostly pertains to heart failure). In order to provide a convincing message and a valuable addition to the literature, the review should be focused on more recent literature.

  1. The manuscript is quite extensive and sometimes off-topic. It is hard to discern the main message from background information. The authors should consider to get to the point and remove information that has been described in other reviews over the past years (e.g., discovery of B vitamins). Molecular functions of most B vitamins have been known for a long time and do not need to be repeated unless novel insights into their functions have been gained recently. Addressing this point is also in line with the previous suggestion.

  1. The writing style and language in general is sometimes hard to understand and often complicates the intended message to some extent. The manuscript also contains several typos that need to be corrected as well. Moreover, the general formulation seems somewhat informal. It is advised to improve this aspect, which will also help convey the message of this review better.

Specific comments:

  1. The figure are nicely made and provide a welcome visualization of what is described in the text. These figures show the molecular interactions with general pathways, but lack any relation to the clinical situation. At least one additional figure would be useful to illustrate how B vitamins could be applied from a translation perspective. The manuscript seems to be an invitation to study B vitamins as therapeutic modalities, but the clinical aspect is missing from the figures. The additional figure could show the translation of the shown molecular mechanism and how patients may benefit.

  1. A table was provided to summarize chapter 6 (B vitamins in heart failure). The same could be done to summarize chapter 4 (energy metabolism and B vitamins) to complement the figures.

  1. Chapter 5 is mostly a repetition of previous reviews and textbooks and could be omitted.

  1. The conclusions drawn in the manuscript merely discuss the caveats of the use of B vitamins, but they could be directed more toward the benefits and obivous potential of B vitamins as a treatment option.

Author Response

Comments and Suggestions for Authors

This review written by Piquereau and colleagues entitled “Metabolic therapy of heart failure: Is there a future for B vitamins?” describes various aspects of heart failure with a focus on energy metabolism. The authors highlight the origins and known molecular functions of all known B vitamins in this lengthy review. Consequently, the authors discuss how each B vitamin might improve cardiac function during heart failure. Ultimately, the authors conclude that clinical studies have not been able to indicate a beneficial effect of B vitamins to date. The manuscript pertains to in-depth discussions of B vitamins and their potentially beneficial function in the cardiovascular system. The authors convincingly demonstrate their expertise on the topic and make a strong case for further studies on B vitamins as a treatment modality. However, some points should be addressed to improve the manuscript.

  1. A principal point of criticism is regarding the novelty of the review in general. Most of the studies described in the review have been discussed extensively in other reviews and the vast majority of the references have not been published in the past 5 years. To illustrate this point: only about 30% of the references were published in 2016 or later (excluding the introduction, which mostly pertains to heart failure). In order to provide a convincing message and a valuable addition to the literature, the review should be focused on more recent literature.

Response: We decided to write this review because we noticed that there was no recent review dealing with this subject. Our idea was to be as exhaustive as possible, and we cited old studies that are part of the history of the potential use of B vitamins in the treatment of cardiovascular diseases. It seems crucial to refer to these studies since they contribute to the overall understanding of this review. Referring the reader to other journals (to which he or she may not have access) would impact the dynamics of the reading and alter our demonstration. Because we tried to cite all the work studying the potential effects of B vitamins in cardiovascular diseases, the studies published in the last 5 years account for 30% of the references. This is actually quite substantial when one considers the number of studies that this represents. The first studies on this subject date back to the early 1990s, so all the studies cited cover a period of about thirty years. The fact that 30% of these studies have been published in the last five years shows the growing interest in B vitamins and citing all the studies highlights this. 

  1. The manuscript is quite extensive and sometimes off-topic. It is hard to discern the main message from background information. The authors should consider to get to the point and remove information that has been described in other reviews over the past years (e.g., discovery of B vitamins). Molecular functions of most B vitamins have been known for a long time and do not need to be repeated unless novel insights into their functions have been gained recently. Addressing this point is also in line with the previous suggestion.

Response: We agree that some information has already been reported elsewhere but this does not mean that it should not be repeated in this manuscript. The understanding of our review requires the reminder of the different functions of B vitamins to highlight the important roles they play in energy metabolism. This is essential to give credit to the idea that they could be used in diseases associated with important disturbances of energy metabolism (such as heart failure). If we systematically referred to other reviews, our demonstration would lose quality and we believe that a manuscript should stand on its own. Furthermore, the reader may not have access to the references, and this would alter the understanding of this review. We choose to write basic information to make sure that there is no hindrance to the understanding of the main message and we want to take advantage of the open-access publication proposed by mdpi to make this information available to the whole scientific community.

  1. The writing style and language in general is sometimes hard to understand and often complicates the intended message to some extent. The manuscript also contains several typos that need to be corrected as well. Moreover, the general formulation seems somewhat informal. It is advised to improve this aspect, which will also help convey the message of this review better.

Response: We agree that the manuscript deserve some editing. We have extensively modified the manuscript by removing non-essential details on energy metabolism regulation and we have shorten sentences that may have been difficult to understand. We corrected typos and the manuscript has been reviewed by an English speaking scientist.

Specific comments:

The figures are nicely made and provide a welcome visualization of what is described in the text. These figures show the molecular interactions with general pathways, but lack any relation to the clinical situation. At least one additional figure would be useful to illustrate how B vitamins could be applied from a translation perspective. The manuscript seems to be an invitation to study B vitamins as therapeutic modalities, but the clinical aspect is missing from the figures. The additional figure could show the translation of the shown molecular mechanism and how patients may benefit.

Response:

 We have added an additional figure which summarizes the role of B vitamins in energy metabolism and their potential benefit in heart failure.

A table was provided to summarize chapter 6 (B vitamins in heart failure). The same could be done to summarize chapter 4 (energy metabolism and B vitamins) to complement the figures.

Response: We agree with this proposal and this is included in figure

  1. Chapter 5 is mostly a repetition of previous reviews and textbooks and could be omitted.

Response: This chapter is only a short review of the main alterations in energy metabolism described in heart failure. We do not go into detail, but it is essential to remind them to understand what B vitamins could bring to heart failure treatment.

The conclusions drawn in the manuscript merely discuss the caveats of the use of B vitamins, but they could be directed more toward the benefits and obvious potential of B vitamins as a treatment option.

Response: We understand the feeling of the reviewer and we have thus directed it on interest of B vitamins in this context.    

Reviewer 3 Report

The current paper provides a comprehensive review of B group vitamins and their potential role in heart failure. My comments are minor and I congratulate the authors for their work.

  • Line 586, presume it is “heart failure” not “failing heart”?
  • I advise to shorten the conclusions.
  • I advise the following citation to be introduced: Mitu, O.; Cirneala, I.A.; Lupsan, A.I.; Iurciuc, M.; Mitu, I.; Dimitriu, D.C.; Costache, A.D.; Petris, A.O.; Costache, I.I. The Effect of Vitamin Supplementation on Subclinical Atherosclerosis in Patients without Manifest Cardiovascular Diseases: Never-ending Hope or Underestimated Effect? Molecules 202025, 1717. https://doi.org/10.3390/molecules25071717

Author Response

Comments and Suggestions for Authors

The current paper provides a comprehensive review of B group vitamins and their potential role in heart failure. My comments are minor and I congratulate the authors for their work.

Line 586, presume it is “heart failure” not “failing heart”?

Response: This has been changed

I advise to shorten the conclusions.

Response: We have shortened the conclusions.

I advise the following citation to be introduced: Mitu, O.; Cirneala, I.A.; Lupsan, A.I.; Iurciuc, M.; Mitu, I.; Dimitriu, D.C.; Costache, A.D.; Petris, A.O.; Costache, I.I. The Effect of Vitamin Supplementation on Subclinical Atherosclerosis in Patients without Manifest Cardiovascular Diseases: Never-ending Hope or Underestimated Effect? Molecules 202025, 1717. https://doi.org/10.3390/molecules25071717

Response: We thank the reviewer for bringing this reference to our attention. This reference has been added in the revised manuscript.

Round 2

Reviewer 2 Report

The authors have addressed specific issues raised regarding the first version. The added figures are a useful addition to the manuscript. While the English edits appear to be done throughout the manuscript, it is hard to determine how extensively this has been corrected without the reviewing a document with Tracked Changes. Overall, my main point of criticism remains regarding the novelty and the length of this review. The authors state that 30% of references published in the last 5 years is a good sign of scientific interest in the field. I tend to disagree with this argument as this is not a reason to write such an extensive review and could have been more limited to recent developments in contrast to present a historical review. However, I will defer this issue to the editorial team as they should decide whether this is within the scope of the journal. The contents are scientifically sound do not warrant a rejection by any extent.

Author Response

We thank the reviewer for his(her) appreciation of our revised version of the manuscript. We apologize for not having provided the track change version but so many were done that we felt it was difficult to read anyway. As regard the comment on the “historical” nature of our review, this is actually our goal to provide in a single review the body of scientific knowledge acquired in this field over the years. We feel this assessment was missing until today. As the reviewer finally states that “the contents are scientifically sound do not warrant a rejection by any extent.” , we sincerely believe that our manuscript is now acceptable for publication.

Round 3

Reviewer 2 Report

Considering the positive comments of the other reviewers and the neutral position of the editorial staff, I can only agree with the author's responses and I believe the current version should be sufficient for publication. Congratulations on this elaborate review in this interesting field.